# Nurturing an organizational context that supports team-based primary mental health care: A grounded theory study

Rachelle Ashcroft [1] *, Matthew Menear[2], Simone Dahrouge[3], Jose Silveira[4], Monica Emode[5], Jocelyn Booton[1], Ravninder Bahniwal[6], Peter Sheffield[1], Kwame McKenzie[4,7]

1 Factor-Inwentash Faculty of Social Work, University of Toronto, Toronto, Ontario, Canada, 2 Faculty of Medicine, Department of Family Medicine and Emergency Medicine, Université Laval, Quebec, Quebec, Canada, 3 Faculty of Medicine, Department of Family Medicine, University of Ottawa, Ottawa, Ontario, Canada, 4 Faculty of Medicine, Department of Psychiatry, University of Toronto, Toronto, Ontario, Canada, 5 School of Population and Public Health, University of British Columbia, Vancouver, Canada, 6 School of Medicine, St. George's University, St. George's, Grenada, 7 Wellesley Institute, Toronto, Ontario, Canada

☉ These authors contributed equally to this work.
* rachelle.ashcroft@utoronto.ca

**Data Availability Statement:** Data cannot be shared publicly because although we have removed identifiable information from transcripts,

## Abstract

### Background

The expansion of the Patient-Centred Medical Home model presents a valuable opportunity to enhance the integration of team-based mental health services in primary care settings, thereby meeting the growing demand for such services. Understanding the organizational context of a Patient-Centred Medical Home is crucial for identifying the facilitators and barriers to integrating mental health care within primary care. The main objective of this paper is to present the findings related to the following research question: "What organizational features shape Family Health Teams' capacity to provide mental health services for depression and anxiety across Ontario, Canada?"

### Methods

Adopting a constructivist grounded theory approach, we conducted interviews with various mental health care providers, and administrators within Ontario's Family Health Teams, in addition to engaging provincial policy informants and community stakeholders. Data analysis involved a team-based approach, including code comparison and labelling, with a dedicated data analysis subcommittee convening monthly to explore coded concepts influencing contextual factors.

### Results

From the 96 interviews conducted, involving 82 participants, key insights emerged on the organizational contextual features considered vital in facilitating team-based mental health care in primary care settings. Five prominent themes were identified: i) mental health explicit in the organizational vision, ii) leadership driving mental health care, iii) developing a mature

we cannot remove all information that could potentially be used to identify participants. Responses to questions contain information describing communities, practice settings, and healthcare providers' personal roles and responsibilities, some of whom live in small rural communities where such information could be used to identify individuals. Data are available from the Associate Dean of Research at the University of Toronto (contact via Dr. David Brennan at david. brennan@utoronto.ca) for researchers who meet the criteria for access to confidential data.

**Funding:** This study was funded by the Canadian Institutes of Health Research (MOP-142435). The funding body had no involvement in the design of the study nor involvement in the collection, analysis, interpretation of data, or writing of the manuscript.

**Competing interests:** The authors declare that they have no competing interests.

and stable team, iv) adequate physical space that facilitates team interaction, and v) electronic medical records to facilitate team communication.

## Conclusions

This study underscores the often-neglected organizational elements that influence primary care teams' capacity to deliver quality mental health care services. It highlights the significance of strong leadership complemented by effective communication and collaboration within teams to enhance their ability to provide mental health care. Strengthening relationships within primary care teams lies at the core of effective healthcare delivery and should be leveraged to improve the integration of mental health care.

## Background

The Patient-Centred Medical Home is an effective model of primary care for treating and coordinating mental health care [1–3]. As a team-based model of primary care, the Patient-Centred Medical Home partners family physicians with a range of interprofessional healthcare providers (IHPs)–such as nurses, social workers, psychologists, pharmacists, and other types of healthcare providers—to enhance comprehensive patient-centered care by broadening access to a range of health and mental health services [2,4]. Some of the guiding principles of the Patient-Centred Medical Home [5] include an emphasis on ongoing patient-provider relationships, whole-person care, accessibility, and comprehensive coordination [6–8]. These guiding principles underpinning the Patient-Centred Medical Home form an optimal structure to deliver timely access to much-needed comprehensive mental health treatment for depression and anxiety [1,2]. Efforts to expand the Patient-Centred Medical Home model continue across the United States and Canada as a way to respond to the shortage of family physicians, and with aims to improve management of chronic diseases, and in some cases, mental health care [1,2,6,7,9]. Despite the aims to strengthen and expand the Patient-Centered Medical Home, research has found inconsistent process and outcomes when it comes to mental health care [10]. It remains unclear, the extent to which mental health care is integrated within the organizational structure of these recently established teams, and how the organizational structure of these teams may help facilitate the delivery of mental health care [1–2,6,7,11].

### Mental health care in primary care

Mental health conditions and substance use disorders were prevalent and leading causes of disability and illness related burden even before the COVID-19 pandemic [12–14]. Since the start of the COVID-19 pandemic, mental health has even worsened with higher rates of depression and anxiety noted across the United States, Canada, and elsewhere [12,15–17]. There are numerous reasons why primary care is well-positioned to address the growing demands for mental health care, particularly for depression and anxiety. Patients struggling with depression and anxiety are common in primary care [18–20], with a significant proportion of mental health care delivered by primary care physicians [21,22]. Treatment for depression and anxiety in primary care is effective, acceptable, and is associated with good patient outcomes [23–25]. Furthermore, there is widespread consensus that strengthening mental health services within primary care is the best way to respond to the high population demands for prevention and management of depression and anxiety [18,26–30]. Historically, however, primary care has

been unable to meet the growing demands for comprehensive mental health care for depression and anxiety [31]. Fortunately, the expansion of the Patient-Centred Medical Home model of primary care provides an opportunity to address the care gap for depression and anxiety by strengthening the integration of team-based mental health services within primary care.

## Organizational context

Organizational theory explains that delivery system changes–such as the integration of mental health care in Patient-Centred Medical Homes–occurs within a broader context that involves dynamic relationships, interactions, and structures that enable and constrain the delivery of care [32–36]. Such contextual factors can support and/or impede efforts to integrate mental health services within primary care [30,31]. According to Valentijin et al. integration is understood as a coordinated approach to the delivery of health and mental health services that are supported by processes at multiple contextual levels in the system including the broader system and policy, and organizational [37].

Alignment of the organizational context of a Patient-Centred Medical Home is needed to support team-based mental healthcare, otherwise efforts to implement team-based mental health care will be unsuccessful [38]. For example, key organizational factors that influence the success of integrating mental health care in Patient-Centered Medical Homes relate to the shared values, beliefs, and activities of the organization [38]. Additionally, Evans et al. identified key organizational capabilities that influenced the success or failure of integrating mental health within primary care [39]. Examples of organizational capabilities include governance, leadership, information technology, and partnerships [39]. Understanding such features of the organizational context is necessary to explain the variations in the delivery of care between primary care practices, and can help provide guidance on strengthening mental health care for depression and anxiety [38,40–43]. According to Evans et al., without understanding the organizational capabilities for integrating mental health care, researchers and leaders will face difficulties in establishing best practices and will likely face unanticipated barriers in the integration of mental health care in team-based settings like Patient-Centred Medical Homes [33].

## Study rationale

As efforts to strengthen the Patient-Centred Medical Home model continue [6,7,11] gaining a better understanding of features within an organizational context that support mental health care is needed [40–43]. To date, less attention has been given to the organizational structures needed to support the integration of mental health in primary care than the technical and clinical aspects of mental health care [44,45]. This study was part of a larger qualitative study investigating the financial and non-financial incentives that influence the quality of mental health care in Family Health Teams [46]–one example of a Patient-Centred Medical Home located in Ontario, Canada. To proceed with our overarching investigation of the financial and non-financial incentives in Family Health Teams [46], we needed to understand the organizational contexts within which these primary care teams delivered mental health care. Recent evidence suggests that variations exist across Family Health Teams in terms of capacity for treating and managing depression and anxiety [47]. One of the explanations for this variation may relate to differences of the organizational contexts across Family Health Teams [37]. This paper presents findings related to the following question of our study: "What organizational features shape Family Health Teams' capacity to provide mental health services for depression and anxiety across Ontario, Canada?". Knowing teams' perspectives on what organizational features

shape Patient-Centred Medical Homes' capacity to deliver mental health care will be critical to inform future policy and practice decisions.

## Methods

### Study design

The design of this study is guided by constructivist grounded theory [48]. Constructivist grounded theory is an inductive research approach that is well suited to guide investigations that encompass multiple viewpoints without imposing pre-existing constructs [48]. Our research team was professionally diverse and included different disciplinary and clinical backgrounds spanning: social work, psychiatry, epidemiology, mental health research, and primary care health services research. In addition, we established an advisory committee consisting of community stakeholders—representing leadership, administration, clinician, and patient perspectives in primary care and/or mental health–whom we consulted throughout the study. Furthermore, grounded theory is considered an ideal method for conducting research aimed at transforming health systems to be more person-centred and comprehensive [49].

### Setting

Canada is a geographically large country comprised of ten provinces and three territories and is sparsely populated with just over 38 million people inhabiting approximately 10 million square kilometers of land (Fig 1). With approximately 18.9% of Canadians residing in rural areas, the majority of Canadians reside in urban settings [50]. With a population of approximately 15 million, Ontario is Canada's most populated province [51].

Family Health Teams are one example of a Patient-Centred Medical Home located in Ontario, Canada [11]. Policy reforms led to the implementation of the first Family Health Team in 2007 [52]. There are now 186 different Family Health Teams operating across Ontario and caring for approximately 22% of Ontarians [53–55]. Although size and team composition vary across the 186 Family Health Teams, they are typically comprised of family physicians, nurses, social workers, pharmacists, dieticians, consulting psychiatrists and others who provide a range of health and mental health services without a burden of direct costs to the patient [56–58].

### Sample population and recruitment

The main sample population for our study consisted of providers and administrators (i.e. executive directors) from the 186 Family Health Teams across Ontario, Canada. Eligible providers and administrators included executive directors, family physicians, nurse practitioners, nurses, and the range of professionals involved in the delivery of mental health care within these Family Health Teams. As per constructivist grounded theory, we used initial and theoretical sampling phases [48]. We strived to include participants representing diverse disciplinary backgrounds, across diverse geographical regions, and from Family Health Teams with variations in team size and composition. In addition, provincial policy informants, as well as some community stakeholders were eligible to participate. We recruited potential participants by sending invitational emails to Family Health Team executive directors and asked that they share the recruitment information with the various professionals working within their Family Health Team. As well, invitational emails were sent to some provincial policy informants, and key community stakeholders. Those who were interested in participating in this study responded by contacting the research coordinator by email.

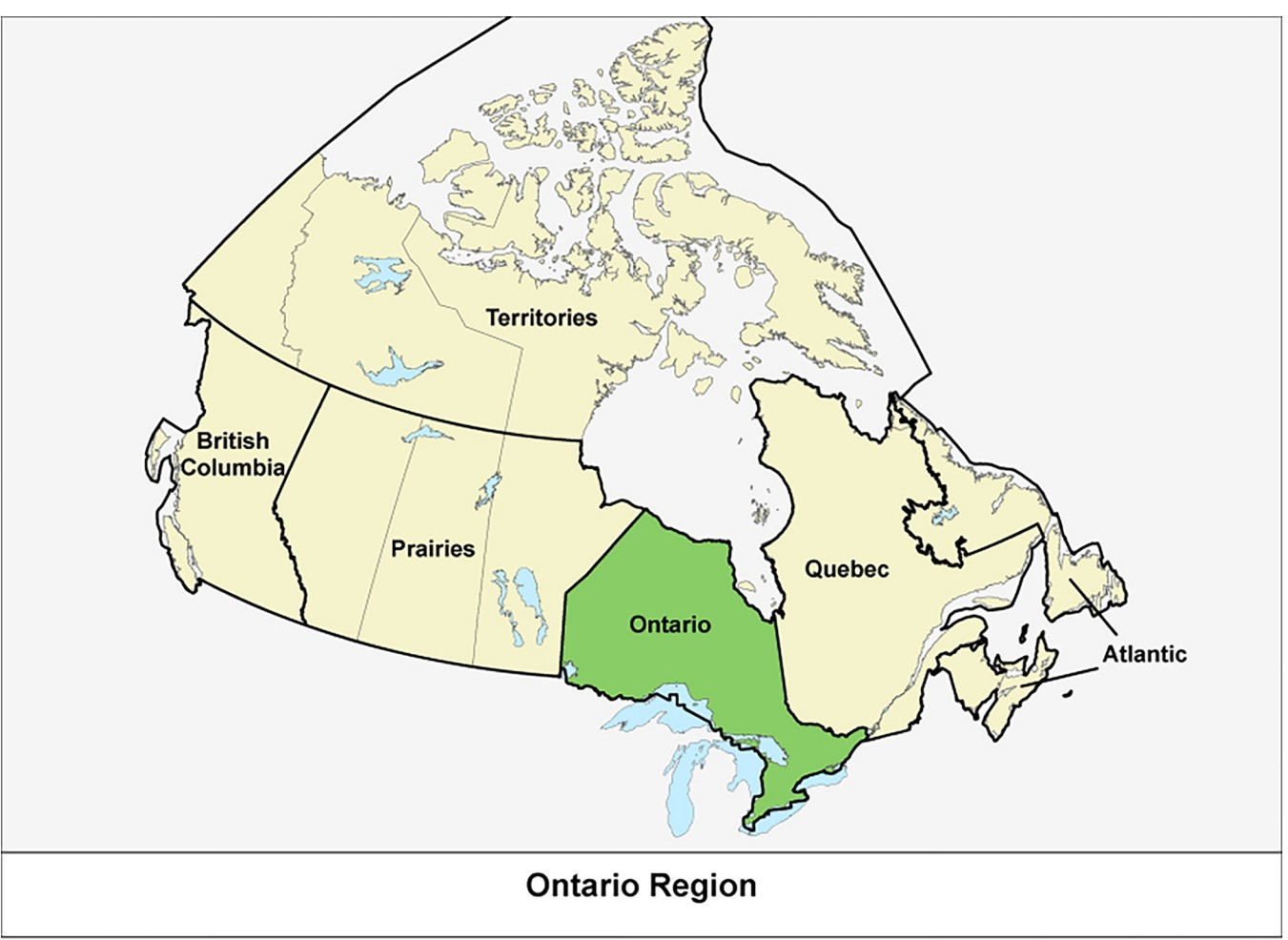

**Fig 1. Map of Canada with Ontario shown in green.**

## Data collection

Consistent with a grounded theory approach, data collection occurred over two phases: i) initial sampling, and ii) theoretical sampling. The initial sampling phase had a descriptive focus, whereby the interviews conducted as part of the theoretical sampling phase aimed to add a deeper explanation into emerging themes. Using a semi-structured interview guide developed by the research team, all one-on-one interviews were conducted in-person by one of two members of the research team (RA/JB) between April 2016 and October 2018 at a location most convenient for the participants. We devised the semi-structured interview guide using sensitizing concepts that helped by providing a starting point, and consistent with grounded theory the interview questions evolved over the course of the two phases of data collection. The sensitizing concepts were derived from our previous research [46]. Interview questions explored organizational features that help and/or hinder the delivery of mental health care for patients with depression and/or anxiety (S1 Table).

Informed written consent was obtained in prior to the conducting of all in-person interviews. With participants' consent, interviews were audio recorded and then transcribed verbatim. Transcripts were returned to each participant to ensure that the transcript was sufficiently de-identified to participants' preferences. After each interview, the interviewer completed

memo-writing. We concluded data collection when we reached saturation at 96 interviews [59]. The reason why so many interviews were necessary to reach data saturation is because this study was part of a larger qualitative study investigating the financial and non-financial incentives that influence the quality of mental health care in Family Health Teams [46]. We are in process of developing a model explaining the financial and non-financial incentives which required an understanding the organizational contexts. Lastly, it required so many interviews to reach saturation because of the commitment to include the range of disciplinary perspectives that comprise this interdisciplinary team-based model of primary care.

### Data analysis

Analysis and data collection occurred in parallel. Consistent with grounded theory, data analysis used initial, focused, and axial coding stages [48]. Initial codes were iteratively developed. During this coding stage, data were first coded using a line-by-line process. Focused coding was conducted to identify between-category relationships [48,60]. Data analysis, code comparison and labelling were completed using a collaborative team-based approach. We created a data analysis subcommittee (RA/MM/JB/ME) to compare coded concepts influencing contextual factors, which met on a monthly basis. In addition, three member-checking meetings with experts in mental health and primary care were held during data analysis. Data analysis continued until data saturation was reached. Trustworthiness was also increased through keeping an audit trail during data collection and analysis [35,41]. Nvivo11 was used to help organize the data analysis process.

### Ethical considerations

Research Ethics Board (REB) approval for this study was obtained from Bruyère Continuing Care (#M16–16–001), Centre for Addiction and Mental Health (CAMH) (#140/2015), St. Joseph's Health Centre/Unity Health Toronto (#15–830), Université Laval (#2016–2877), University of Toronto (#33175), and the University of Waterloo (#20973). Each email invitation sent to potential participants included information about study aims. Informed consent was obtained prior to the conducting of all in-person interviews. Participants were provided their de-identified interview transcript to ensure that the transcript was de-identified to their satisfaction.

### Results

We conducted 96 interviews with 82 participants. Of the 82 participants involved in this study, we interviewed 14 participants twice–once in the initial sampling phase of data collection, and a second time in the theoretical phase of data collection. Of the N = 82 participants, $n = 65$ were providers and administrators working in Family Health Teams, $n = 9$ were policy informants, and $n = 8$ were key community stakeholders. See Table 1 for an overview of the professional roles of participants. There were $n = 18$ Family Health Teams represented in our sample, with $n = 11$ located in urban settings and $n = 7$ located in rural settings situated across all six of Ontario health regions (North East, North West, East, Central, Toronto, and West).

Participants provided insights on the organizational contextual features they considered important to facilitate team-based mental health care in primary care settings such as Patient-Centred Medical Homes. We identified five themes in the data: i) mental health explicit in the organizational vision, ii) leadership driving mental health care, iii) developing a mature and stable team, iv) adequate physical space that facilitates team interaction, and v) electronic medical records to facilitate team communication.

Table 1.  Professional roles of participants (N = 82).

| Professional Roles | |
|---|---|
| **Family Health Team Participants: Providers and Administrators** | **n = 66** |
| Social Work* | 14 |
| Family Physician | 11 |
| Executive Director | 10 |
| Mental Health Counsellor | 9 |
| Psychiatrist | 7 |
| System Navigator | 3 |
| Nurse | 2 |
| Nurse Practitioner | 2 |
| Occupational Therapist | 2 |
| Program Manager* | 2 |
| Psychologist | 2 |
| Outreach Worker | 1 |
| Pharmacist | 1 |
| **Policy Informants** | **n = 9** |
| **Key Community Stakeholders** | **n = 8** |
| **TOTAL** | **83*** |

*One participant held two part-time roles which is why professional role total is 83 for N = 82.

## Mental health care explicit in the organizational vision

Most participants spoke about the importance of having mental health explicit in the organizational strategic plan. Participants emphasized the importance of having a common team vision and strategic plan for team-based mental health care that was explicitly communicated in the form of guiding principles, mission and/or vision statements. For example, an executive director elaborated:

> *Early on, we developed guiding principles and essentially those were the guiding principles. . .we believe in the shared care model. . .so basically if you don't share that vision, you don't belong with [us]. So, it also helps us in our recruitment now. . .it's important when we bring people now into the organization that they believe the same thing. We can't have people who won't function in a team, and so if people don't work with the culture, then they probably do need to work somewhere else, because you just can't have people who don't believe in those principles because it just skews everything* (P225, executive director).

A program director from another Family Health Team agreed and stated, "*priorities are important at [the] start, then it just flows*" (P111, manager). A family physician also explained that having mental health in the organizational mission statement helps guide decision-making, "*We really do come back to the mission a lot, in a lot of the statements and the work and the decisions that are made. . .It's our mission to see and provide primary care to patients, to a population with mental illness*" (P117, family physician). Participants explained that having a common team vision guided physicians' commitments to seeing patients for mental health. For example, an executive director noted that, "*Our mission is to provide primary care to people with serious mental illness*" and continued by stating that as a testament of the organizational mission "*Each physician has to commit to take up to 20% of the practice to be people with serious*

*mental illness"* (P114, executive director). Additionally, many participants highlighted that a core value of their primary care organization was to respond to the needs of their patients and community. *"[We] really hold on to those grassroots community values which is, we are here to serve the community"* (P119, social worker).

Contrarily, some participants noted that not having mental health explicitly embedded in organizational principles, mission and/or value statements meant that mental health was not a priority for patient care. For example, an executive director explained, *"Everything's very focused. . .on the medical aspect of it. . .everything but mental health"* (P128, executive director). As well, there were a few participants who indicated that they were unaware as to how organization's priorities, mission and value statements were used to guide mental health care. A mental health counsellor noted, *"Honestly, I don't know what our mission statement says. . .That's an exercise you sit down and do as a group then you never look at it again"* (P101, mental health counsellor).

Overwhelmingly, participants spoke about the importance of the team having a shared commitment to collaborative mental health care. The team commitment to mental health care, as described by most participants, was in-part due to the organizational culture that prioritized mental health. A mental health counsellor explained, *"Everybody that I work with in this Family Health Team believes that mental health is a real thing and they really understand the connection between physical health and wellness, and mental health and wellness, and that the two will influence each other"* (P107, mental health counsellor). This mental health counsellor went on to explain that the team's shared understanding of the importance of mental health enhanced patient care: *"The [family physicians are] really proactive in picking up on something. . .because they caught it early you are able to do more preventative stuff. . .Which isn't necessarily true in other medical settings"* (P107, mental health counsellor). Similarly, a family physician highlighted the commitment to mental health across all team members:

> *The whole team is very sensitive and compassionate and recognizes that we're going to see a population. . .from the front staff to the nursing staff. . .everyone's very familiar with working with the population of people with mental illness. . .I think they're very good at it and the interactions are positive and understanding, from physicians right through to the front desk and everyone else* (P117, family physician).

Interestingly, an executive director noted that the shared commitment across the team to mental health care was because it was necessary to respond to patient needs:

> *We invested a fair chunk of resource into mental health and addictions as a Family Health Team. . .they had no idea how much their day would be focused on mental health. When I ask a family doctor, How much of your day is focused on mental health?, they said. . .Probably 30 percent. . .certainly anxiety, depression, mood disorders and stuff like that, 30 percent of basically any primary care provider's day, safe to say. So there was a strong reason for us to invest, strong motivation among the providers to engage* (P237, executive director).

## Leadership driving mental health care

Formal and informal leadership was identified as important drivers of mental health care within the Family Health Teams. Such champions were considered essential for establishing the organizational culture, with one participant stating *"[Leadership] is kind of more about a philosophy of care, and I think it's a culture-based issue set often from the top"* (P231, community informant). Many participants explained the importance of having family physicians

championing mental health, as explained by a psychiatrist: "*The lead physicians have always been major advocates. If you don't have that, nothing's going to work*" (P228, psychiatrist).

Several participants noted that the importance of the clinical governance structure for driving mental health care. Some participants noted that the community-led governance model of their Family Health Team helped focus their organization on mental health care. An executive director noted the importance of the community providing guidance to their Family Health Team by way of their Board of Directors: "*Having a community board gives us direct access to patients, because when our community board members are out in the community, they are speaking to people, so we always have that tie*" (P121, executive director).

## Developing a mature and stable team

Developing a mature and stable team was achieved by hiring the right composition of providers, funding the right roles, recruitment and retention, and unique rural challenges. Participants emphasized the importance of developing a strong team which starts with hiring the right composition of interprofessional providers. According to a family physician: "*We're also careful about the roles we've asked to fund within our Family Health Team*" (P116, family physician). Teams described themselves as adapting the roles they funded based on need: "*We had two social workers. When one went off on maternity leave we hired a nurse system navigator instead*" (P113, occupational therapist). Compositions, however, were impacted by resources being linked to roster size: "*We don't actually have enough patients to normally warrant a full-time mental health worker. They just shifted funds from maybe nursing support into hiring that person*" (P213, family physician). Some team leaders explicitly described allocating resources strategically to produce desired compositions for providing mental health care: "*Our social worker [is] the program coordinator for [our mental health program]. If I were to ask, could we have that position, it would have been no, but why would I ask and why would the Ministry care. If you're delivering a service, you want to deliver it as effectively as you can*" (P237, executive director). Participants explained, however, that establishing the appropriate composition of teams and hiring the right types of providers for mental health can be challenging without guidelines: "*There's not been kind of enough discussion on like, how do we utilize, or how do we structure our team to make sure it meets this need?*" (P118, social worker). Hiring the right people went beyond looking at an individual's skill set: "*We all had previous relationships with them at the hospital or at [community mental health agency. . .So they kind of handpicked people that they really trust*" (P147, mental health counselor).

While some teams described the hiring for mental health professionals as fluid–"*It was really smooth. We had a lot of really good qualified social workers*" (P208, mental health nurse)–several participants described difficulties pertaining to recruitment and retention: "*We get enough funding. We can't find the body. . . It's recruitment!*" (P219, executive director). Similarly, a social worker from a different Family Health Team noted "*I'm probably the third [social worker in this position in] maybe five years*" (P249, social worker). According to a policy informant, difficulties with recruitment and retention is disruptive to the team: "*If you constantly have people who are leaving, it is going to impact on the level of understanding, and the cohesiveness of teams*" (P243, policy informant). There were some participants, however, who indicated that their Family Health Team had no difficulties with recruitment or retention: "*Anyone who came at the start, almost everyone who came at the start, is still with us*" (P237, executive director). Some participants explained that Family Health Teams located in rural regions had unique challenges with recruitment and retention because of the lack of supply of human resources, and as a result, had to be more flexible in hiring. An executive director explained the challenges she faced in hiring social workers for her Family Health Team: "*In a small town,*

*you don't necessarily have to be a social worker or have a social work degree. You can have a degree in another background and work as a social worker. You need to fill the position, so you'll take whatever might be closest to that position to do the job"* (P128, executive director).

## Adequate physical space that facilitates interaction

Across interviews, participants explained that good relationships and communication were necessary for team-based primary mental health care. Many participants emphasized the importance of adequate physical space to enable practitioners to be co-located and on-site simultaneously. As noted by a social worker, *"One of the things that helps is just the set up in the clinic, we are all located in the same area"* (P112, social worker). Physical space facilitated opportunities for direct communication, as described by a mental health counsellor: *"There is less chance for miscommunication. We share office space. . . I really think that we are way more effective"* (P107, mental health counsellor). A psychiatrist also elaborated that physical space provided the setting and team co-location was important, yet the degree to which team members interacted went beyond just the physical space, *"That's partly where the co-location part plays in, but even that can be not enough. You can be co-located and have your door closed all the time! Or you can have it open and loiter and talk in spaces when you have the time"* (P211, psychiatrist). Participants explained that fostering the desired collaborative interaction between providers was difficult, despite being co-located, when teams were comprised of provider working part-time and full-time. A program manager noted, *"A lot of our positions are part-time which makes things even more difficult when you're trying to plan interprofessional care"* (P111, program manager). The result *"[w]e don't talk together because we are not usually here at the same time so that makes it difficult"* (P108, nurse practitioner).

Participants identified challenges that emerged, however, when the physical space was inadequate to accommodate co-location of all members of the team. For example, some participants indicated that the number of team members exceeded the capacity of available space, which created conflict that deterred collaboration. A Family Health Team social worker explained, *"Fighting for rooms and that sort of thing makes it difficult [and] it makes it difficult for people to connect with each other"* (P149, social worker). Physical space limitations also created barriers to expanding the team, despite the demands for mental health care: *"How are we supposed to hire more people without having that infrastructure of where to put people?"* (P123, social worker). In some Family Health Teams, inadequate space limited the possibility of holding group-based mental health services with patients as described by a nurse practitioner: *"We've wanted to hold some. . .group sessions. . .we have a board room in the back, but it's not big enough for people"* (P108, nurse practitioner).

Some providers belonged to larger multi-site organizations whereby the IHPs were not physically co-located with physicians, which some participants explained was challenging for the delivery of mental health care. Some participants explained that not being physically co-located meant that variations in care processes existed across the sites which created some challenges for the mental health providers on the team: *"The process is a little bit different in each office because we've tried to gear the service to meet the needs of each individual family doctor's office, which has its pros and cons"* (P103, social worker).

## Electronic Medical Records (EMRs) to facilitate team communication

Most participants highlighted the importance of EMRs to facilitate communication between the various providers. An executive director explained that EMRs provided team members with a site to document and communicate to the team about up-to-date patient progress and goals: *"[We have] documentation on what their goals are so the other team members are*

*supporting that. Using the EMR, every single interaction with a patient has a documentation and so the team can go in and see 'Oh yeah this is the goal'"* (P212, executive director). Participants stated that EMR communication was frequent and important to patient care, particularly for large sites: *"It's not just the physical proximity. . .the virtual proximity allows a lot of docs who are seeing someone and just don't know what to do, to shoot me a message, it's a pretty easy template for being able to make a referral and then go from there. So that's helped a great deal"* (P141, community informant).

## Discussion

Our study provides qualitative insights into the organizational contextual features of team-based primary care practices that influence their capacity to integrate mental health care. Although there is existing research on organizational structures in primary care [61–64], our study directly addresses the often-neglected structural dimensions of primary care models as it pertains to mental health care [53,64–66] and has implications for both the design of new primary care organizations and the improvement of existing teams. Ample literature exists that focuses on the bolstering of individual provider capacity by way of education and training to improve the treatment and management of mental disorders in primary care, yet little research exists that specifically focuses on the role of the organizational context in shaping the prioritization of mental health care in team-based primary care settings.

### Strategic planning

This study demonstrates the importance of explicitly integrating mental health care in a primary care's organizational strategic plan and vision. Commitment to collaborative mental health care can be strengthened by engaging team members in developing a shared team vision and direction that can guide organizational and program-level decision-making [67]. Institutional theory developed by Meyer and Royan has a longstanding recognition of the importance of nurturing organizational culture, values, and behaviors through the use of such strategic planning [68]. In primary care teams, organizational-level agreement on practice goals among staff may even help promote interprofessional collaboration during difficult challenges and conflicts [69]. The challenge, however, is that there have been limited guidelines related to the inclusion of mental health care within primary care organizations' strategic planning, which has led to inconsistencies and vast variations across primary care organizations [53,66,70]. One way to foster a culture that nurtures mental health is by having a range of interprofessional perspectives contributing to organizational processes, such as priority setting.

### Leadership

This study highlights the importance of formal and informal leadership that prioritizes and drives the integration of mental health care within and across the primary care team. Leadership refers to having the ability to nurture an organizational context whereby all employees can contribute to their fullest potential in a way that supports the organization's mission and vision [71]. Leadership in healthcare contexts typically includes the ability to identify and shape priorities, and foster a collective commitment across multiple sectors and/or providers to address those priorities [72,73]. Relational aspects of leadership include communicating regularly and clearly across various levels of an organization, acknowledging diverse opinions, sharing knowledge, and engaging meaningfully with others [74]. In this way, leadership is integral for the buy-in across all providers on the importance of mental health care, and consistently aligning priorities and decisions with the organizational aim of mental health care. During many interviews, primary care providers overwhelmingly spoke about the important

role that their executive director played in their organization by keeping mental health care a priority, and by acting as a conduit across the various providers–particularly in larger multi-site Family Health Teams. Transformational leadership has historically recognized the role of leadership in nurturing value-driven behavior by relaying the shared morals and values across the organization [75].

Formal leadership in the form of organizational governance is an important foundation to the integration of mental health in primary care teams [46,76], as was noted in this study. Clinical governance facilitates the adherence of clinical principals, organizational priorities, and standards. Yet there is a paucity of research examining clinical governance as it pertains to the integration of mental health in primary care [45]. Findings from this study suggests that clinical governance is an integral organizational structure that can facilitate the uptake of mental health as a priority in interprofessional primary care teams. While effective leadership continuity and mental health champions can foster a culture that favours mental health, there are variations across primary care teams in terms of expertise and capacity for mental health care [48]. For teams without mental health champions, ongoing support available outside the practice may be used to develop stronger commitments to mental health care within the practice. We believe that other primary care teams and umbrella organizations with existing resources or implementation strategies should be leveraged [77].

## Developing a mature and stable team

Prioritizing mental health care within a primary care team requires a sufficient number and type of providers who have the skills and competencies to identify, treat, and coordinate mental disorders [78]. Findings in this study emphasize the necessity of developing a mature and stable team comprised of the right composition of interprofessional providers who are in the right roles and maximizing their scopes of practice in relation to mental health care. Although interprofessional primary care teams are considered integral for community-based mental health care [2], there is a paucity of research to guide organizational-level decision-making on the right composition of providers–and how to optimize their combined scopes of practice— to meet the demands for mental health care.

There is a widespread health human resources crisis that is expected to worsen in the coming years due to trends such as an aging population, a shift away from comprehensive family practice and burnout among primary care providers [79]. Since the onset of the COVID-19 pandemic, there has been a spike in mental health care demands, which along with the health human resources crisis, highlights the immediate need to attract and retain mental health professionals in primary care [47,80]. A systematic review conducted by Rahim et al. found that a shortage of trained health professionals—and difficulties with recruitment and retention—significantly affected the extent to which mental health was integrated into primary care [45]. Rural and urban primary care teams are facing various challenges in the human resources crisis. Geospacing analysis has demonstrated that there is an unbalanced distribution of primary care providers across the urban-rural continuum which created gaps in access even prior to the immediate crisis at hand [81]. In urban settings, however, coordinated interventions and intersectoral collaboration are needed across institutions are needed to build up the urban environment as a component of the solution [82].

## Providing opportunities for relationships and communication

Findings from our study highlighted the importance of co-location to enable team members to share physical space which facilitated relationship building and communication. Consistent with the growing literature [83–89], findings in our study demonstrate that physical co-

location is an asset for team-based mental health care because it facilitates knowledge-sharing, consultation, relationship building, and collaboration. Similarly, EMRs were considered an essential mechanism to enable communication across team members. Aligned with existing literature, we recommend teams create frequent opportunities, whether brief or informal, for interprofessional communication to enhance collaborative patient-centred care, particularly for those with complex mental health care needs [68]. Experimenting with novel technological means for their facilitation, such as "virtual warm hand-offs" [90], may also assist in the cultivation of these processes for teams distributed due to scale or rurality. The data collection for our study was completed prior to the COVID-19 pandemic when most teams were still working in-person on-site, and prior to the widespread shift of using virtual care in primary care teams [47]. Research is needed to determine optimal methods to promote relationships and collaboration of interprofessional primary care teams in current organizational contexts that include on-site, virtual, and hybrid methods of doing mental health care. Even so, the results from our study highlight the importance of relationship building between providers regardless of physical and electronic structures. Given today's elusive workflows, we predict that improving relationship building, communication, coordination, informality, and knowledge sharing between providers is even more crucial for effective mental health care delivery.

## Limitations

We conducted this study focusing on one model of primary care in Ontario, Canada which may not represent all models of team-based primary care across different geographies.

## Conclusion

Our study aimed to provide a deeper understanding of the organizational features that shape Family Health Teams' capacity to provide mental health services for depression and anxiety. The five themes identified in the data were as follows: i) mental health explicit in the organizational vision, ii) leadership driving mental health care, iii) developing a mature and stable team, iv) adequate physical space that facilitates team interaction, and v) EMRs to facilitate team communication. We believe that nurturing trusting relationships within teams is at the heart of primary care and should be prioritized as human resources and mental health crises worsen in the post-pandemic climate.

## Supporting information

**S1 Checklist. Consolidated criteria for reporting qualitative studies (COREQ): 32-item checklist.**
(PDF)

**S1 Table. Interview guide.**
(DOCX)

**S1 File.**
(PDF)

**S2 File.**
(PDF)

**S3 File.**
(PDF)

**S4 File.**
(PDF)

**S5 File.**
(MSG)

**S6 File.**
(PDF)

**S7 File.**
(PDF)

## Acknowledgments

Thank you to all the participants who took time out of their busy schedules to participate in this study.

## Author Contributions

**Conceptualization:** Rachelle Ashcroft, Matthew Menear, Simone Dahrouge, Jose Silveira, Kwame McKenzie.

**Data curation:** Rachelle Ashcroft, Matthew Menear, Monica Emode, Jocelyn Booton.

**Formal analysis:** Rachelle Ashcroft, Matthew Menear, Simone Dahrouge, Jose Silveira, Monica Emode, Jocelyn Booton, Peter Sheffield, Kwame McKenzie.

**Funding acquisition:** Rachelle Ashcroft, Matthew Menear, Simone Dahrouge, Jose Silveira, Kwame McKenzie.

**Investigation:** Rachelle Ashcroft, Matthew Menear, Monica Emode, Jocelyn Booton.

**Methodology:** Rachelle Ashcroft, Matthew Menear, Simone Dahrouge, Jose Silveira, Monica Emode, Jocelyn Booton, Kwame McKenzie.

**Project administration:** Rachelle Ashcroft, Monica Emode, Jocelyn Booton.

**Resources:** Rachelle Ashcroft, Matthew Menear.

**Software:** Rachelle Ashcroft.

**Supervision:** Rachelle Ashcroft.

**Writing – original draft:** Rachelle Ashcroft, Matthew Menear.

**Writing – review & editing:** Rachelle Ashcroft, Matthew Menear, Simone Dahrouge, Jose Silveira, Monica Emode, Jocelyn Booton, Ravninder Bahniwal, Peter Sheffield, Kwame McKenzie.

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
