## [Decision Letter · Decision Letter 0]

1 Dec 2023

PONE-D-23-28612Nurturing an Organizational Context that Supports Team-Based Primary Mental Health Care: A Grounded Theory StudyPLOS ONE

Dear Dr. Ashcroft,

Thank you for submitting your manuscript to PLOS ONE. After careful consideration, we feel that it has merit but does not fully meet PLOS ONE’s publication criteria as it currently stands. Therefore, we invite you to submit a revised version of the manuscript that addresses the points raised during the review process.

Please note that we have only been able to secure a single reviewer to assess your manuscript. We are issuing a decision on your manuscript at this point to prevent further delays in the evaluation of your manuscript. Please be aware that the editor who handles your revised manuscript might find it necessary to invite additional reviewers to assess this work once the revised manuscript is submitted. However, we will aim to proceed on the basis of this single review if possible. 

The reviewer has raised a number of minor concerns. They feel that the manuscript would benefit from additional details in the methods section, specifically to make the supplementary data available. They also feel that it would be useful to include a more thorough description of the results of the grounded theory phase. 

Could you please carefully revise the manuscript to address all comments raised?

We look forward to receiving your revised manuscript.

Kind regards,

Johanna Pruller, Ph.D.

Associate Editor

PLOS ONE

2. Please ensure that you have specified a) Did participants provide their written or verbal informed consent to participate in this study?

Reviewers' comments:

Reviewer's Responses to Questions

**Comments to the Author**

1. Is the manuscript technically sound, and do the data support the conclusions?

Reviewer #1: Yes

2. Has the statistical analysis been performed appropriately and rigorously? 

Reviewer #1: N/A

3. Have the authors made all data underlying the findings in their manuscript fully available?

Reviewer #1: No

4. Is the manuscript presented in an intelligible fashion and written in standard English?

Reviewer #1: Yes

5. Review Comments to the Author

Reviewer #1: Dear author,

First of all, I enjoyed reading your manuscript. Overall it is a scientifically sound article. You present interesting results that may also help to improve primary mental health care in other countries. Nevertheless, I have a few questions (minor issues):

1. You describe a mixed methods approach. During the first phase grounded theory was applied. Following that a semistructured interview was used. My impression is that the results from the grounded theory phase are not extensively described. Is it possible to add a short paragraph on these results (a couple of sentences)?

2. To interpret the results it would help if the semistructured interview (or the topic list) was available as supplementary material. Is it possible to make this available (I might have overlooked)?

3. It surprises me that so many interviews were necessary to reach data saturation. Can you elaborate a bit on this?

4. I am not familiar with the Canadian health care system. I do know that distances in your country tend to be large, with large urban areas and more remote rural communities. I imagine that needs and availability of mental health care services may differ for these areas. Indeed, you reflect a bit on what is necessary in rural areas. Can you also reflect a bit on the needs in cities (were also teams from Toronto included?)? Do mental health care teams also hold conferences or have other meetings where they meet colleagues from other teams (as this also helps to learn from each other and improve the way care is delivered)?

5. It might be interesting to add a map of Ontario with the locations of the teams that were included in the study.

Thank you for the opportunity to read your interesting work!

6. PLOS authors have the option to publish the peer review history of their article (what does this mean?). If published, this will include your full peer review and any attached files.

Reviewer #1: **Yes: **Klaas Huijbregts

---

## [Author Response · Author response to Decision Letter 0]

15 Jan 2024

Dr. Johanna Pruller

Editor-in-Chief

PLOS One

January 02, 2024

Dear Dr. Pruller:

We are pleased to submit revisions to our manuscript titled “Nurturing an Organizational Context that Supports Team-Based Primary Mental Health Care: A Grounded Theory Study”. We appreciate the editor and reviewer’s feedback and have revised the manuscript in a way that responds to all editor and reviewer comments. Please find below an itemized response to all reviewer and editor comments: 

Reviewer Comment:

First of all, I enjoyed reading your manuscript. Overall it is a scientifically sound article. You present interesting results that may also help to improve primary mental health care in other countries. 

Response:

Thank you, we appreciate your encouraging comments and also appreciate the time you took to review our manuscript. 

Reviewer Comment:

You describe a mixed methods approach. During the first phase grounded theory was applied. Following that a semi-structured interview was used. My impression is that the results from the grounded theory phase are not extensively described. Is it possible to add a short paragraph on these results (a couple of sentences)? 

Response: 

We added mention of grounded theory in the methods and reorganized the data collection section to enhance clarity about the grounded theory methods. 

The following was added to the Data Collection section: 

“We devised the semi-structured interview guide using sensitizing concepts that helped by providing a starting point, and consistent with grounded theory the interview questions evolved over the course of the two phases of data collection. The sensitizing concepts were derived from our previous research [46]. Interview questions explored organizational features that help and/or hinder the delivery of mental health care for patients with depression and/or anxiety (S1).”

Reviewer Comment:

To interpret the results it would help if the semistructured interview (or the topic list) was available as supplementary material. Is it possible to make this available (I might have overlooked)?

Response: 

Interview questions are now added as a supplementary file. The following sentence was added to the data collection section: 

“Interview questions explored organizational features that help and/or hinder the delivery of mental health care for patients with depression and/or anxiety (S1).”

Reviewer Comment:

It surprises me that so many interviews were necessary to reach data saturation. Can you elaborate a bit on this? 

Response: 

Thank you for raising this. We added the following to the Data Collection section to provide clarity: 

“The reason why so many interviews were necessary to reach data saturation is because this study was part of a larger qualitative study investigating the financial and non-financial incentives that influence the quality of mental health care in Family Health Teams [46]. We are in process of developing a model explaining the financial and non-financial incentives which required an understanding the organizational contexts. Lastly, it required so many interviews to reach saturation because of the commitment to include the range of disciplinary perspectives that comprise this interdisciplinary team-based model of primary care.”

Reviewer Comment:

I am not familiar with the Canadian health care system. I do know that distances in your country tend to be large, with large urban areas and more remote rural communities. I imagine that needs and availability of mental health care services may differ for these areas. Indeed, you reflect a bit on what is necessary in rural areas. Can you also reflect a bit on the needs in cities (were also teams from Toronto included?)? Do mental health care teams also hold conferences or have other meetings where they meet colleagues from other teams (as this also helps to learn from each other and improve the way care is delivered)? 

Response: 

We added the following sentences in Methods under the Setting sub-heading: 

“Canada is a geographically large country comprised of ten provinces and three territories, and is sparsely populated with just over 38 million people inhabiting approximately 10 million square kilometers of land (Figure 1). With approximately 18.9% of Canadians residing in rural areas, the majority of Canadians reside in urban settings.” Additionally, we added a map of Canada that illustrates Ontario. 

Also, the following is now added to the results section: “There were n = 18 Family Health Teams represented in our sample, with n = 11 located in urban settings and n = 7 located in rural settings situated across all six of Ontario health regions (North East, North West, East, Central, Toronto, and West).”

The following was also added to the discussion section: 

“Rural and urban primary care teams are facing various challenges in the human resources crisis. Geospacing analysis has demonstrated that there is an unbalanced distribution of primary care providers across the urban-rural continuum which created gaps in access even prior to the immediate crisis at hand [81]. In urban settings, however, coordinated interventions and intersectoral collaboration are needed across institutions are needed to build up the urban environment as a component of the solution [82].”

Reviewer Comment: 

It might be interesting to add a map of Ontario with the locations of the teams that were included in the study. 

Response: 

We added a map of Canada that highlights Ontario. Although we agree it would be helpful to illustrate the locations of the teams, we did not add the locations to the map because it would then identify the study participants. Instead, we added the following sentence to the results section: 

“There were n = 18 Family Health Teams represented in our sample, with n = 11 located in urban settings and n = 7 located in rural settings situated across all six of Ontario health regions (North East, North West, East, Central, Toronto, and West).”

Editor Comment:

Response: 

The manuscript was revised throughout to meet the PLOS ONE’s style requirements including headings, references, and title page. 

Editor Comment: Please ensure that you have specified a) Did participants provide their written or verbal informed consent to participate in this study?

Response: The following sentence was added to Data Collection section: “Informed written consent was obtained in prior to the conducting of all in-person interviews.”

Editor Comment: 

In your Data Availability statement, you have not specified where the minimal data set underlying the results described in your manuscript can be found. PLOS defines a study's minimal data set as the underlying data used to reach the conclusions drawn in the manuscript and any additional data required to replicate the reported study findings in their entirety. All PLOS journals require that the minimal data set be made fully available. For more information about our data policy, please see http://journals.plos.org/plosone/s/data-availability.

Response: 

We are unable to publicly share our data set because although we have removed identifiable information from transcripts, we cannot remove all information that could potentially be used to identify participants. Responses to questions contain information describing Family Health Team communities, practice settings, and providers’ personal roles and responsibilities, some of whom live in small rural communities where such information could be used to identify individuals. Upon reasonable request, the corresponding author is able provide de-identified data. 

Editor Comment: 

Your ethics statement should only appear in the Methods section of your manuscript. If your ethics statement is written in any section besides the Methods, please delete it from any other section. 

Response: 

The following statement is located under a subheading titled "Ethical Considerations" in the Methods section of the manuscript: 

“Research Ethics Board (REB) approval for this study was obtained from Bruyère Continuing Care (#M16–16-001), the Centre for Addiction and Mental Health (CAMH) (#140/2015), St. Joseph’s Health Centre/Unity Health Toronto (#15–830), the University of Waterloo (#20973), the University of Toronto (#33175), and Université Laval (#2016–2877).”

Editor Comment: 

Response: 

The reference list was reviewed and revised. No retracted papers currently appear in the reference list.

---

## [Decision Letter · Decision Letter 1]

25 Mar 2024

Nurturing an Organizational Context that Supports Team-Based Primary Mental Health Care: A Grounded Theory Study

PONE-D-23-28612R1

Dear Dr. Ashcroft,

We’re pleased to inform you that your manuscript has been judged scientifically suitable for publication and will be formally accepted for publication once it meets all outstanding technical requirements.

Kind regards,

Nabeel Al-Yateem, PhD

Academic Editor

PLOS ONE

Additional Editor Comments (optional):

Reviewers' comments:

Reviewer's Responses to Questions

**Comments to the Author**

1. If the authors have adequately addressed your comments raised in a previous round of review and you feel that this manuscript is now acceptable for publication, you may indicate that here to bypass the “Comments to the Author” section, enter your conflict of interest statement in the “Confidential to Editor” section, and submit your "Accept" recommendation.

Reviewer #1: All comments have been addressed

2. Is the manuscript technically sound, and do the data support the conclusions?

Reviewer #1: (No Response)

3. Has the statistical analysis been performed appropriately and rigorously? 

Reviewer #1: (No Response)

4. Have the authors made all data underlying the findings in their manuscript fully available?

Reviewer #1: (No Response)

5. Is the manuscript presented in an intelligible fashion and written in standard English?

Reviewer #1: (No Response)

6. Review Comments to the Author

Reviewer #1: (No Response)

7. PLOS authors have the option to publish the peer review history of their article (what does this mean?). If published, this will include your full peer review and any attached files.

Reviewer #1: **Yes: **Klaas Huijbregts

---

## [Editor Report · Acceptance letter]

4 Apr 2024

PONE-D-23-28612R1 

PLOS ONE

Dear Dr. Ashcroft, 

I'm pleased to inform you that your manuscript has been deemed suitable for publication in PLOS ONE. Congratulations! Your manuscript is now being handed over to our production team.

Kind regards, 

on behalf of

Dr. Nabeel Al-Yateem 

Academic Editor

PLOS ONE